# Autonomous Planetary Liquid Sampler (APLS) for *In Situ* Sample Acquisition and Handling from Liquid Environments

**DOI:** 10.3390/s24186107

**Published:** 2024-09-21

**Authors:** Miracle Israel Nazarious, Leonie Becker, Maria-Paz Zorzano, Javier Martin-Torres

**Affiliations:** 1School of Geosciences, University of Aberdeen, Meston Building, Aberdeen AB24 3UE, UK; javier.martin-torres@abdn.ac.uk; 2Institute of Space Systems (IRS), University of Stuttgart, 70569 Stuttgart, Germany; leonie_becker@posteo.de; 3Centro de Astrobiología (CSIC-INTA), Torrejon de Ardoz, 28850 Madrid, Spain; zorzanomm@cab.inta-csic.es; 4Instituto Andaluz de Ciencias de la Tierra (CSIC-UGR), 18100 Granada, Spain

**Keywords:** liquid environment, instrumentation, sample acquisition, autonomy, reusability

## Abstract

Many natural and artificial liquid environments, such as rivers, oceans, lakes, water storage tanks, aquariums, and urban water distribution systems, are difficult to access. As a result, technology is needed to enable autonomous liquid sampling to monitor water quality and ecosystems. Existing in situ sample acquisition and handling systems for liquid environments are currently limited to a single use and are semi-autonomous, relying on an operator. Liquid sampling systems should be robust and light and withstand long-term operation in remote locations. The system components involved in liquid sampling should be sterilisable to ensure reusability. Here, we introduce a prototype of a liquid sampler that can be used in various liquid environments and may be valuable for the scientific characterisation of different natural, remote, and planetary settings. The Autonomous Planetary Liquid Sampler (APLS) is equipped with pre-programmed, fully autonomous extraction, cleaning, and sterilisation functionalities. It can operate in temperatures between −10 °C and 60 °C and pressure of up to 0.24 MPa (~24 m depth below mean sea level on Earth). As part of the control experiment, we demonstrate its safe and robust autonomous operation in a laboratory environment using a liquid media with *Bacillus subtilis*. A typical sampling procedure required 28 s to extract 250 mL of liquid, 5 s to fill the MilliQ water, 25 s for circulation within the system for cleaning and disposal, and 200 s to raise the system temperature from ~30 °C ambient laboratory temperature to 150 °C. The temperature is then maintained for another 3.2 h to sterilise the critical parts, allowing a setup reset for a new experiment. In the future, the liquid sampler will be combined with various existing analytical instruments to characterise the liquid solution and enable the autonomous, systematic monitoring of liquid environments on Earth.

## 1. Introduction

Water quality monitoring effectively avoids environmental risks and promotes sustainable operation and surveillance of different artificial water infrastructure [1]. Furthermore, water sampling and characterisation are also relevant to investigate different ecological environments, such as oceans, rivers, natural pools, aquaculture sites such as fisheries, water distribution infrastructure in cities, etc. [2,3,4,5,6]. Various parameters can be monitored in a natural or artificial liquid ecosystem, such as the quantity and nature of organic matter in the water sample, pH, salinity, presence and nature of microorganisms, concentration of different dissolved gases, etc. [7]. However, all these independent studies require a preliminary stage where a liquid sample is acquired.

Because of the inherent difficulties of working in this environment, scientific instruments must be durable, affordable, transportable, and fully automated. Given the existing stresses and operation conditions in liquid environments, any deployed technology must comply with stringent mass, volume, and power consumption limits. Liquid samplers should be reusable, and as microbial communities can proliferate in these media, biasing the measurements, the critical parts of the sampling system should be sterilisable. Here, we describe the Autonomous Planetary Liquid Sampler (APLS), which can collect multiple large-volume samples of liquid, operating with liquid temperatures between −10 °C and 60 °C and pressures up to 0.24 MPa (24 m depth below mean sea level on Earth). The primary goal of the APLS is to allow the chemical and microbiological monitoring of natural remote ecosystems on our planet. To avoid cross-contamination between repeated sample collections, the instrument shall be sterilisable. Once collected, the samples can be further processed to characterise their physical and chemical properties and the nature of the microbial community using different instruments [8].

This instrument shall collect pristine samples autonomously from brine pools and cold springs from the Arctic and Antarctic regions for its future application to planetary exploration. There is also interest in the long-term monitoring of oceans, measuring variables such as temperature, pressure, salinity, density, tracers, nutrients, isotopes, noble gases, etc., to answer some of the questions about ocean–atmosphere and ocean–ice interactions [9]. The study of eDNA (or environmental DNA) is another exciting prospect in this research that can significantly contribute to characterising marine biodiversity and opening new pathways to monitoring it [10]. 

Sampling in liquid media can be even more challenging in some natural extreme environments, such as deep in the ocean or hydrothermal environments. Some existing prototypes use shape memory alloy, which senses high temperatures and actuates the suction mechanism to sample liquids hotter than 80 °C [11]. In contrast, others use a gas-tight isobaric sampler [12,13,14] or a torsion bar to generate the energy required to seal the sample chambers [15]. Some instruments have been designed to collect large-volume liquids at high temperatures and depths [16]. In contrast, others use an osmotically driven continuous liquid sampler [17], based on a flow-through and pressure-tight liquid sampler [18], or use a solid phase microextraction sampler [19], including a serial sampler capable of acquiring multiple deep-sea, gas-tight liquid samples [20]. Finally, more recent solutions include a pressure-tight sampler with a metal seal capable of acquiring high-purity liquid samples [21] or Keika Ventures’ Kynar^®^ bags [22,23]. Among these techniques for sampling extreme liquid environments, using a pressure-tight sampler to control and regulate the influx of liquid and avoid leakages has proven to be the most reliable solution at higher operating pressures. However, the lack of cleaning and sterilisation functions means that the existing in situ sample acquisition and handling systems for liquid environments are currently limited to single use. The semi-autonomous nature of these samplers, relying on an operator, also imposes heavy financial and time constraints on sampling missions. These shortcomings can be overcome with a reusable sampler such as APLS.

Furthermore, for the samplers mentioned above to be adaptable for a wide range of planetary applications, they must be reusable without needing periodic maintenance. One currently proposed development for future planetary sampling applications is the IceMole, which incorporates a sterilisation function with a 3% hydrogen peroxide (H_2_O_2_) solution [24]. These design solutions have been tested in similar environments on Earth, such as subglacial brine pools within the cold glacier of Blood Falls, McMurdo Dry Valleys, Antarctica [25] and the cold springs in the Canadian Arctic [26,27], which have clear biological potential.

This paper outlines the design choices and proposed methods for extracting, cleaning, and sterilising of the sampling unit. We demonstrated its utility with a pilot prototype tested in the laboratory with a *Bacillus subtilis* culture. A Simulink model was also developed to support the design and testing phases.

## 2. Materials and Methods

### 2.1. Functional Requirements

The main driver for the liquid sampler design was its capability to minimize environmental impact to support the search for life at different ocean depths and in liquid environments such as rivers, natural pools, artificial pools for fisheries, water distribution infrastructure in cities, etc. Repetitive sampling from different locations or consecutive periods will require cleaning and sterilisation between samples to avoid cross-contamination. The material choice for the systems (especially the tubes) is essential. All parts need heat and pressure resistance and a protective coating to avoid corrosion by salty, acidic, or alkaline waters. Also, the materials selected must be compatible with a pressure-tight system that uses a pressurized noble gas (like helium). Furthermore, utilising and disposing of the cleaning solvent is another challenge. This section addresses these issues and discusses the extraction, cleaning, and sterilisation design choices.

### 2.2. Extraction

An advanced flow control system is needed to extract and move the liquid samples in precise quantities. This requires a combination of pressure manifolds, valves, and pumps. For the APLS design, considerations for pump selection included lifespan, flow rate, discharge pressure, high precision and reproducibility, compatibility with cleaning and sterilisation options, low-temperature operation, heat resistance during sterilisation, chemical resistance against brines and liquids with particles, self-priming, bidirectionality (to aid one pump for both extraction and cleaning processes if necessary), energy efficiency, and low wear and maintenance. From previous published work, we have estimated the amount of liquid required for biological analysis using DNA extraction and nanopore sequencing to be 300 mL [28]. This amount set the baseline requirement of the sampling time and amount that became the driving factor for the pumping system selection among peristaltic, piston, rotary vane, and gear/screw/lobe pumps.

Piston or rotary vane pumps would be the best option for sampling in ocean environments. Hydraulic shocks or water hammers are prevalent at ocean depths with pressures up to ~108.6 MPa. The choice of pump needs to decrease the impact and velocity of these shocks. A flexible impeller rotary vane pump is a good option because it can withstand hydraulic shocks better than fixed components. Gear and screw pumps were omitted due to their problems of wear and poor efficiency. We chose a piston pump because it can operate in a wide temperature, pressure, and viscosity range, and with better energy efficiency. 

Furthermore, this pump type can build pumping pressure faster and deliver high precision sampling. Since the current design does not require backward movement of the liquid, we did not consider the bidirectionality of the pump. The cleaning solvent storage is connected with the pump’s natural direction. This will be discussed in detail in the next section. However, the tubing needs to be flexible and durable to adapt to the vast pressure differences, and the materials used need to endure high pressure and temperatures between −10 °C (sampling liquid) and 150 °C (sterilisation temperature), including a ±10 °C margin. Plastic tubing could be a good choice since it offers a vast range of operating temperatures. In addition to the pump, the extraction system would require an inlet filter to remove particles above 1 mm in diameter, a safety pressure regulation system, and relief valves. Also, the sensors for monitoring temperature, pressure, and flow must be chosen to withstand high pressures. A peristaltic pump might be better and cheaper for a prototype made only for liquid sampling in non-arctic environments on Earth. However, that design choice would require significant changes to adjust to colder liquids. 

### 2.3. Cleaning

The choice of solvent was the main criterion for the cleaning process. Several alternatives were considered, such as polar protic solvents (water, alcohol), polar aprotic solvents (acetone, acetate), and non-polar solvents (chloroform, diethyl ether, toluene, hexane). Since most chlorides and iodides are insoluble in water, polar solutes will dissolve better in polar solvents, and non-polar solutes (e.g., hydrocarbons) dissolve better in non-polar solvents, the design choice was confined to the ocean environment, which is predominantly brine. Also, the fact that solubility generally increases with heat was a deciding factor in the design to utilise the heating system of the sterilisation process to aid better dissolution of salts and rinsing with water through the tubes repeatedly to remove residual salts and particles from the internal walls of the pump and other components. 

MilliQ water, distilled water, alcohols, and detergents were considered for cleaning solvents. MilliQ water was chosen as the ideal solvent because it has good solubility for salts and other particles that can be carried away through the water movement into a wastewater container or disposed of.

### 2.4. Sterilisation

The goal was to choose a simple sterilisation and cleaning method that is effective against microorganisms, does not add much mass, poses no hazard, includes easy disposal/storage of residues, and has good cleaning properties, low operation time, and low power consumption requirements. 

Several sterilisation options are considered in planetary exploration and in space applications, such as (1) heating to at least 150 °C for 3.2 h (at 4 to 6 orders of magnitude bioburden reduction) according to the European Cooperation for Space Standardization (ECSS) standards for dry heat bioburden reduction for flight hardware (ECSS-Q-ST-70-57C); (2) alternatively, the usage of chemicals such as H_2_O_2_, ultraviolet (UV), and gamma rays can also be considered. Heating the whole system from cold operating temperatures to 150 °C would be power-consuming and could damage heat-sensitive materials such as electronics. However, other chemical options (ethylene oxide gas, nitrogen dioxide gas, ozone, hydrogen/vapourised hydrogen peroxide, peracetic acid, glutaraldehyde, and formaldehyde) are more complex due to safety hazards in the event of leakage. Ultraviolet or gamma rays are also not a viable option since the sources of these radiation forms are not portable, need significant power, and cannot operate efficiently in liquid environments. Finally, the dry heat sterilisation method was chosen for its simplicity, portability, and long-space heritage. 

### 2.5. Other Design Requirements

Cleaning and sterilisation protocols require that all the parts in contact with the liquid be decontaminated and thermal-, chemical-, and ageing-resistant materials be used for tubes, pump parts, valves, etc. For instance, when sampling from acidic or corrosive liquids, a resistive or anti-corrosive coating would be required. Heating from cold temperatures to 150 °C requires a high-power supply and sound insulation. Also, to demonstrate the sampler in outer space conditions, the prototype must fit the dimensions of the SpaceQ chamber (300 mm × 300 mm × 220 mm), which is a space and Mars simulation environment [29]. The electronic components and any tethers for delivering power and transmitting data must be protected against liquid and the environment (see Table 1). Since after repeated sampling operations there is a high chance of clogging the internal parts of the sampler, there must be an inbuilt detection system. For example, flow sensors can detect clogging of the sampler by monitoring drastic reduction in the flow parameter. 

When the inlet valve is opened while the sampler is submerged in the high-pressure environment, the resultant hydraulic shock or water hammer can result in erosion, ruptures, and failure of tubes, valves, pumps, etc. Therefore, safety and blowoff valves are required to prevent pressure spikes and to regulate system pressure. Other ways to suppress this effect would be by using hydraulic dampers, expansion tanks, non-slam check valves, opening and closing the valves slowly, avoiding elbows or loop pipes, and using flexible pipe materials.

### 2.6. Simulink Modelling

A Simulink model of the APLS system for extraction, cleaning, and sterilisation was built to facilitate the application of different design choices for pumps, valves, etc., and to specify the location of the heater mats for efficient sterilisation (see Figure 1). The models were quickly adapted for several operation scenarios by changing the components’ parameters. Upon optimisation of the final model parameters, the components for the proof-of-concept prototype were selected. 

### 2.7. APLS Schematic and Proof-of-Concept Prototype

Figure 2 shows the conceptual design of the APLS system and the prototype, which includes a positive displacement pump with predominantly plastic working elements, P (diaphragm pump), for extracting the liquid sample and circulating the MilliQ water for cleaning operation. The current prototype used several COTS (commercial off-the-shelf) components. We used a 12 V diaphragm pump (Flojet RLFP122202A, RS PRO, London, UK) with 3.8 LPM, self-priming of up to 0.8 m, and operating pressure of up to 2.4 bar (or 0.24 MPa) as a low-cost alternative. A total of 24 V 3-Way solenoid valves (ASCO SCG356B466VMS, RS PRO, Manchester, UK) were used at the inlet (V1) and for sample 1 (V2), sample 2 (V3), wastewater (V4), and MilliQ water (V5) for controlling liquid flow. The inlet temperature is measured with an inline temperature sensor, T1 (10 K NTC thermistor, Alphacool, London, UK), and system temperature and flow are measured with an inline temperature/flow sensor, TF1 (10 K NTC thermistor, Alphacool, UK), at the cleaning loop between V5-V1-V2-V3-V4-V5. This temperature/flow sensor gives an accurate temperature and flow rate reading for each extraction, where the flow rate can be converted to a volumetric reading using elapsed time since the pump was switched on. The prototype used ¼ in. stainless steel pipes (316 SS) between all components and ¼ in. silicone flexible tubing at the inlet. A total of 2 × 60 W silicone heater mats were used along the stainless steel pipes to heat them to the required temperature during the sterilisation operation. At the end of the inlet tube, a copper filter with a mesh size of 1 mm in diameter (Atlas Scientific, Brooklyn, NY, USA) was fitted to prevent large particles from entering the system. The material choice of copper was to limit microorganisms growing and clogging the filters and, subsequently, the interior of the pipes in the system [30]. We also used 250 mL polypropylene (PP) bottles (RS PRO, London, UK) for sample, wastewater, and MilliQ water storage. For switching the pump, valves, and heater, an 8-channel relay module was controlled by an ESP32S3 Dev Board with WiFi capability that can transmit temperature, flow, and pump/valve/heater status data remotely. The circuit diagram, ESP32S3 programmable codes, and remote data visualization tools using Grafana are provided in the Appendix A.

### 2.8. Cleaning and Sterilisation Validation of APLS with Bacillus subtilis Culture

To demonstrate the cleaning and sterilisation efficiency of the APLS, we performed culture-based analysis (colony forming units, CFU) of samples passed through the system; this process is summarised in the scheme of Figure 3. The procedures highlighted by orange blocks were conducted in a Laminar Air Flow cabinet to minimize ambient microbiological contamination. A 10 mL solution of *Bacillus subtilis* (NCIMB 3610, Aberdeen, UK) culture (~O.D600 = 1) was sampled through the inlet of the APLS by running the pump for 2 s. This was collected in a sterile 15 mL Falcon tube (Fischer Scientific, Loughborough, UK). After this sampling stage, MilliQ water was circulated and flushed through the system for 200 s, followed by the dry heat microbial reduction (DHMR) procedure at 150 °C for 10,000 s. After this sterilisation, a new sample of 10 mL of clean MilliQ water was acquired with the APLS and collected in another sterile 15 mL Falcon tube (Fischer Scientific, Loughborough, UK). The Bacillus subtilis samples were diluted to a factor of 10^6^. Three replicates were taken for culture-based analysis for both *Bacillus subtilis* and MilliQ samples. In each replicate, 100 µL of the solution was plated in nutrient agar plates (Merck, Gillingham, UK). CFU counts were calculated from the number of colonies formed after 24 h of incubation at 30 °C.

## 3. Results and Discussion

The experimental setup for testing the APLS proof-of-concept prototype was run in the laboratory, but the instrument was operated autonomously. The data were logged in the InfluxDB Cloud database (https://eu-central-1-1.aws.cloud2.influxdata.com (accessed on 9 September 2024)) and plotted using Grafana (https://www.grafana.com (accessed on 9 September 2024)) for real-time visualization and debugging. The instrument is designed so that, by installing a communication unit, all the data can be viewed remotely when operating autonomously.

Here, we elaborate on the prototype’s performance against the functional design requirements. The tests demonstrated the system’s extraction, cleaning, and sterilisation aspects. A typical sampling procedure required 28 s for extracting 250 mL of liquid into the sample bottle, 5 s for filling the MilliQ water, 25 s for circulation within the system for cleaning and disposal, and 200 s for raising the temperature of the system from ~30 °C ambient laboratory temperature to 150 °C, which was maintained for another 3.2 h for sterilisation at four to six orders of magnitude of bioburden reduction. 

### 3.1. Extraction

Figure 4 shows the extraction procedure for 28 s for 250 mL of liquid into the sample 1 bottle at a maximum flow rate of 10.21 mL/s (average = 8.946 mL/s). The inlet tip temperature sensor monitors the liquid body’s temperature, which helps avoid extracting liquid from temperatures below or above a threshold that could damage the tubing and internal systems. Suppose the temperature is within the desirable range, typically between −10 °C and 60 °C for the prototype. In this case, the inlet and sample 1 valves would be opened to create a clear way between the liquid body and the sample 1 bottle. At this point, the pump was turned on, and the liquid was extracted for 28 s until the sample 1 bottle was packed to its limit (250 mL). 

Figure 4 also shows the extraction performance of the APLS as predicted by the model. The pump was tuned to match the liquid output of the prototype and yielded a maximum flow rate of 8.958 mL/s. The model assumes stable flow rates, so for most of the extraction time, the flow rate was the same. At the start of the pump operation, the system experienced a backflow, which is nominal if the system pressure is higher than the liquid body pressure. But this can be arrested by using a check valve depending on the desirable direction of flow.

### 3.2. Cleaning

Figure 5 shows the cleaning procedure using MilliQ water. This procedure comprises three steps: filling, circulation, and flushing. Filling the operational part of the tubing circuit takes about 5 s. The MilliQ valve was opened for this period, and the pump was turned on. Note that the inlet and disposal valves are closed at this point. Once the MilliQ water was filled in the system (44.73 mL, calculated based on the average flow rate of the pump), the pump was turned on again for 25 s to circulate the MilliQ water, which helps dissolve any contaminants and residues. Closing the inlet valve also helps create a loop within the system to facilitate circulation. After this, the disposal valve was opened while the pump was working to flush out all the filling liquid into the wastewater bottle.

Figure 5 also shows the cleaning liquid filling performance of the APLS as predicted by the model. For a given pump flow rate (8.958 mL/s), the model predicts that 56.59 mL of MilliQ water would be filled in the system if the pump was run for 5 s. This is still within a safe limit since the total internal volume of the pipes is 65.814 mL. Note that the initial burst in the flow rate is due to the elevation difference of the MilliQ water bottle, which was set to be at 13 cm higher than the pump to allow gravity-assisted feed.

### 3.3. Sterilisation

Figure 6 shows the sterilisation procedure using the heater mats. The system has two heater mats equally distributed throughout the length of the tubing. Supplying current through the mats resulted in homogenous conductive heating of the tubing. A temperature difference of ΔT = ~120 °C was achieved during 200 s to raise the temperature to 150 °C. After this, the temperature must be maintained for the 3.2 h required for the dry heat microbial reduction (DHMR) procedure outlined by the ECSS standards. This has been achieved by incorporating PID temperature control. The current prototype did not have a separate heater for the inlet tip, but the residual convective heat was responsible for raising the temperature of the inlet tip by about 10 °C. 

Figure 6 also shows the main difference in the performance of the model in comparison to the prototype. Since the current prototype does not have inbuilt temperature sensors at various parts of the pipe, achieving a reliable average temperature was difficult. But the model results show the thermal discrepancies throughout the system, helping to devise a method to achieve an average temperature that complies with ECSS standards for DHMR procedures. Typically, temperatures of the parts closest to the heat source were higher than the rest of the system. Also, the material properties of different parts of the pipes influenced the heat distribution. Due to this, the model overinterpreted the thermal distribution of the system, resulting in 312 s to achieve an average temperature of 150.1 °C compared to 200 s for the prototype.

### 3.4. Mass, Power, and Data Budget

As expected, the sterilisation procedure had the highest power demand, up to 50 W (160 Wh), due to the enormous temperature difference required (ΔT = ~120 °C). However, the power budget of a typical sampling cycle was not much more than this value, making it reasonable to operate. The current prototype used several COTS (commercial off-the-shelf) components that added to its size and mass, as well as the system’s thermal mass. Several parameters were monitored simultaneously for a seamless system operation, including continuous housekeeping data for inlet tip temperature, average pipe temperature, and statuses of the pump, heater, and valves at 1 Hz. During the individual operation of the system, data on the flow rate and volume, along with the individual status of the pump, heater, and valves, were transmitted. For 11 variables of data types *int* and *float*, each 4 bytes in size, a total of 510 KB of data were generated for a sampling cycle, including extraction, cleaning, and sterilisation, as mentioned in Table 2.

### 3.5. Cleaning and Sterilisation Validation of APLS with Bacillus subtilis Culture

CFU counting of the agar plates was performed after 24 h; see Figure 7. Experiments (a) and (c) represent the positive and negative controls. Figure 7a is the resulting culture agar plate for cultivating the diluted *Bacillus subtilis* liquid sample acquired by the APLS system. This indicates multiple colonies; each is produced after 24 h by replicating one single initial cell. Figure 7b is the resulting culture agar plate for the cultivation of the MilliQ solution that was circulated through the APLS system after the sterilisation cycle. This indicates that there are no colonies, and thus no viable cells in the system after cleaning and sterilisation at high temperatures. A similar test was run with fresh MilliQ water directly plated on the agar plate, showing no colonies (see Figure 7c).

The results shown in Figure 6 and Table 3 demonstrate no contamination between samples and the effectiveness of the APLS system’s autonomous sterilisation procedure. 

### 3.6. Limitations and Potential Solutions

This section addresses some of the issues encountered during the design and testing of the APLS prototype. (i) Corrosion: most of the parts of the sampler used stainless steel pipes, which after sustained contact with liquids such as brines can lead to corrosion in the inner walls of the pipes. These corrosive particles could contaminate the sampled liquid and potentially damage the sensors, valves, and pump. Therefore, for a long-term application, a protective anti-corrosion coating may be desirable. (ii) Clogging: during our design and testing phases, we did not encounter this issue by employing a copper inlet filter, but prolonged use may finally result in clogging. If a filter is used in a sample where microorganisms can proliferate, this may also result in clogging. Filter clogging can result in failure of the pumps. (iii) Undesirable heating of critical parts: since the APLS prototype includes bulk heating of all the pipes using heater mats, it is not easy to maintain strict temperature limits for certain critical parts of the system. For example, the nominal upper temperature limit of the pump was 60 °C, but due to the lack of localised heating in the current version, the valves and pump could be heated over the limit, affecting its performance. The immediate solution to this would be to use a pump with metallic working elements, which have a higher temperature rating. (iv) Internal pressure buildup: lack of pressure relief valves in the current version of the prototype means that the pressure from pumping the liquid is contained within the system and released only when the disposal valve is opened. Though the pressure increase is within the tolerance limits of the system, it is not desirable to operate the system at unstable conditions for prolonged periods.

## 4. Conclusions

In this paper, we presented a pilot prototype of the Autonomous Planetary Liquid Sampler (APLS) with pre-programmed autonomous sampling, cleaning, and sterilisation functionalities for liquid bodies with temperatures between −10 °C and 60 °C and pressures of up to 0.24 MPa (~24 m depth below mean sea level on Earth). We demonstrated its safe and robust operation in a laboratory environment. We have modelled the system using Simulink for extraction, cleaning, and sterilisation procedures. Such samplers can be helpful for sample extraction and handling in laboratory setups, as demonstrated during ocean missions and by distributing the raw/processed samples to different downstream analytical instruments. Further work and improving of the Technological Readiness Level (TRL) during high-pressure and high-temperature operation could be vital for APLS to be considered a possibility in upcoming missions to ocean worlds such as Europa, Enceladus, Titan, etc. Meanwhile, the sampler is projected to be useful for Earth’s environmental and ocean science communities. 

## Figures and Tables

**Figure 1 sensors-24-06107-f001:**
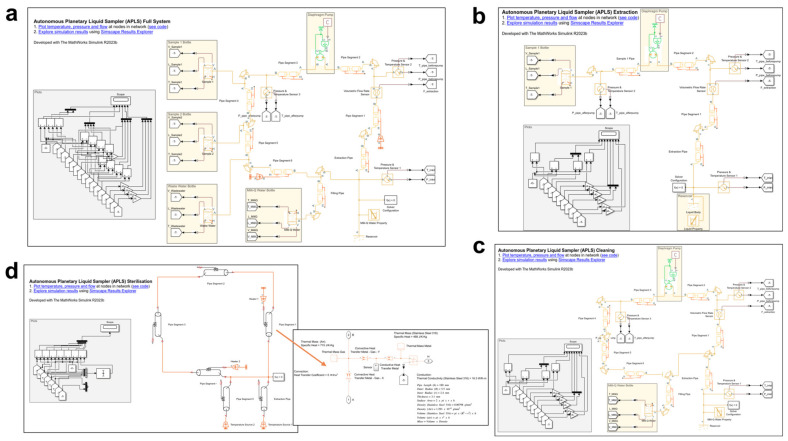
APLS Simulink models of the (**a**) full system, (**b**) extraction, (**c**) cleaning, and (**d**) sterilisation, with the thermal circuit within the pipe highlighted by an arrow at the inset figure. The underlined blue characters will allow to explore the simulation results and code from the model in one click. The uppercase characters represent ports of different components of the thermal liquid model (A and B for inlet and outlet of the pipe segments respectively, H for heater, C for constant, R for rotational motion, T for temperature, L for level, and V for volume of the bottle).

**Figure 2 sensors-24-06107-f002:**
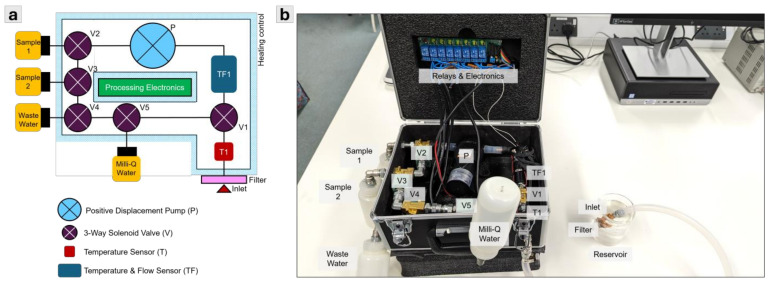
(**a**) Schematic of the APLS concept; (**b**) APLS proof-of-concept prototype.

**Figure 3 sensors-24-06107-f003:**
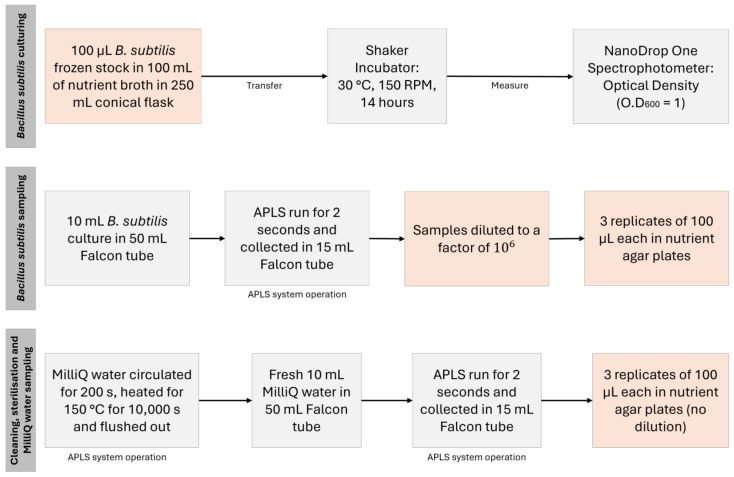
Summary of the cleaning and sterilisation validation experiment with *Bacillus subtilis* culture. The procedures highlighted by orange blocks were conducted in a Laminar Air Flow cabinet to minimize ambient contamination.

**Figure 4 sensors-24-06107-f004:**
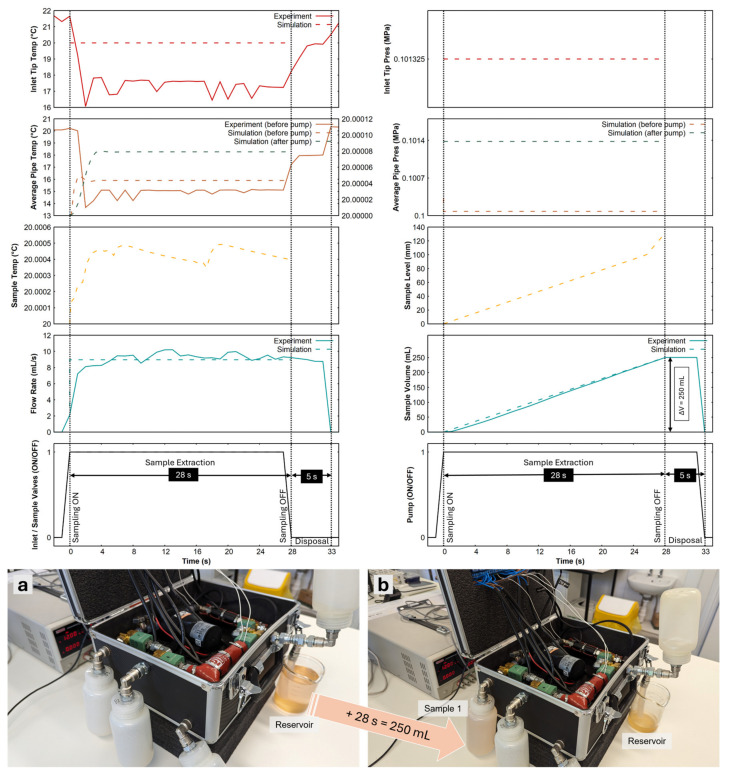
Sample extraction of 250 mL of water at 8.946 mL/s in ambient laboratory conditions. Plots of inlet tip and average pipe temperatures, inlet tip and average pipe pressures, sample temperature, sample level, flow rate, volume and inlet status, sample valves, and pump are shown. The experimental results are shown as solid lines and simulations as dashed lines (measurements are on the right y-axis for some subplots). The model predicted a maximum flow rate of 8.958 mL/s to achieve an extraction volume of 250.8 mL in 28 s in ambient laboratory conditions with a liquid temperature of 20 °C. (**a**) Before sample extraction from the reservoir glass beaker; (**b**) after Sample 1 was extracted in 28 s.

**Figure 5 sensors-24-06107-f005:**
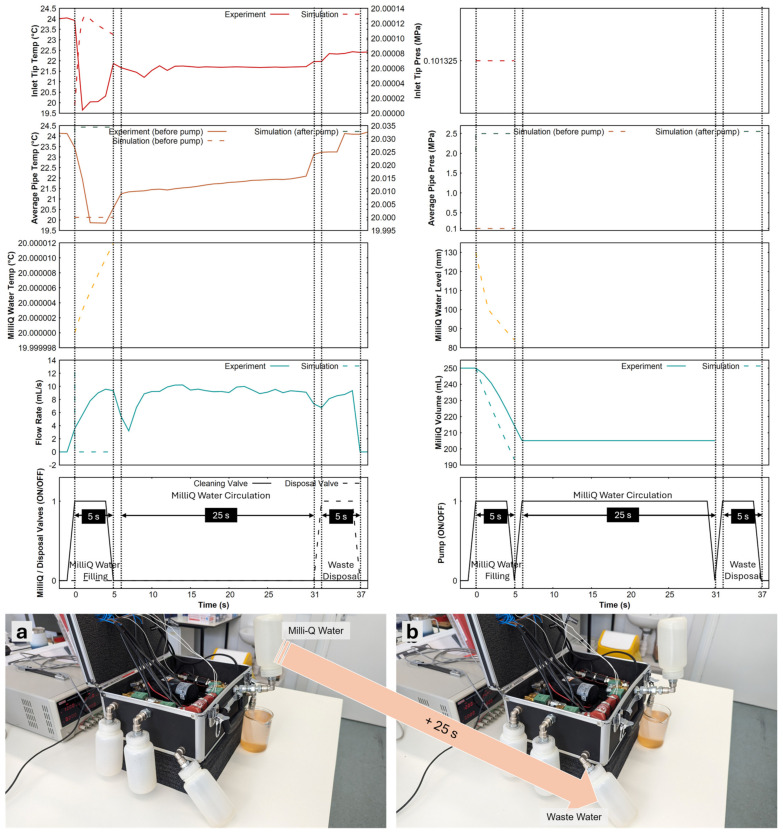
Cleaning procedure with MilliQ water in ambient laboratory conditions with a liquid temperature of 20 °C. Plots of inlet tip and average pipe temperatures, inlet tip and average pipe pressures, MilliQ water temperature, MilliQ water level, flow rate, volume and status of MilliQ water, disposal valves, and pump are shown. (**a**) About 44.73 mL of MilliQ water was extracted into the system (5 s of pump operation) and (**b**) circulated for 25 s for cleaning and disposal into the wastewater bottle. The model predicts 56.59 mL of MilliQ water was extracted into the system during the 5 s of pump operation. The experimental results are shown as solid lines and the simulation as dashed lines (measurements are on the right y-axis for some subplots).

**Figure 6 sensors-24-06107-f006:**
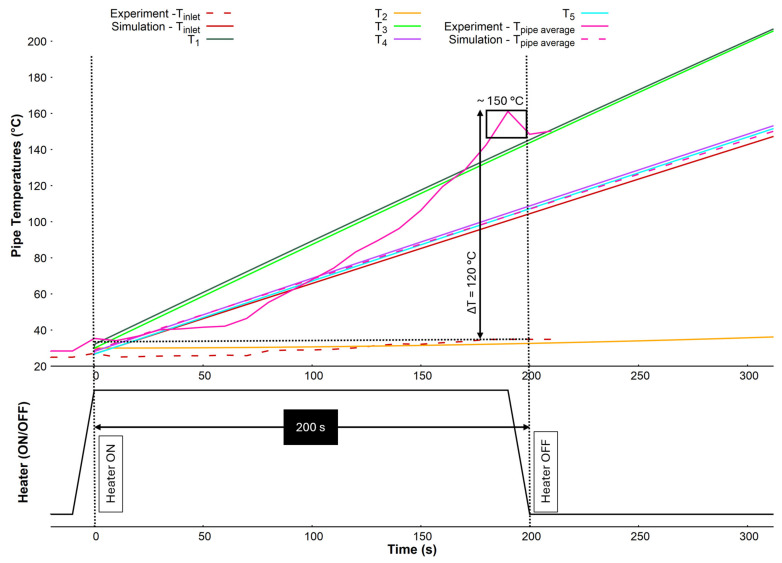
Inlet tip temperature (T_inlet_) and average pipe temperature (T_pipe average_) during the 200 s heating cycle demonstrated a steady increase in average pipe temperature to ~150 °C in ambient laboratory conditions (ΔT = ~120 °C). Model-predicted inlet tip temperature, temperatures of pipe segments 1, 2, 3, 4, 5 (T_1_, T_2_, T_3_, T_4_, T_5_), and average pipe temperature are also shown. The model predicted a 312 s heating, demonstrating a steady increase in average pipe temperature to 150.1 °C in ambient laboratory conditions (ΔT = ~120 °C). The experimental results are shown as solid lines and simulation results as dashed lines.

**Figure 7 sensors-24-06107-f007:**
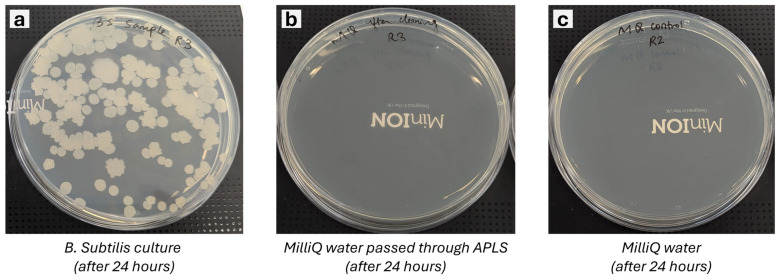
Culture plates after 24 h of incubation of (**a**) *Bacillus subtilis* culture, (**b**) MilliQ water passed through APLS after *Bacillus subtilis*, and (**c**) MilliQ water as control.

**Table 1 sensors-24-06107-t001:** Temperature properties of the electronics and their monitoring.

Parameter	Properties
Temperature Range of Electronics	Conventional-temperature electronics: −55 °C/−65 °C to 125 °C Low-temperature electronics: −150 °C to −273 °C
Temperature Measurement Range: PT1000	−70 to 550 °C, ±(0.3 + 0.005 × t) °C accuracy
Temperature Measurement Range: Low-Temperature Sensor	−200 °C to 0 °C, ±0.5 °C calibrated accuracy

**Table 2 sensors-24-06107-t002:** APLS system budgets.

Parameter	Value
Dimensions	300 mm × 250 mm × 150 mm (Width × Depth × Height)
Mass	2 kg
Power	Extraction: 18.371 W (0.143 Wh) Cleaning: 14.796 W (0.103 Wh) Sterilisation: 50 W (160 Wh) Typical budget per sampling cycle: 83.167 W (160.246 Wh)
Data Products and Type	Inlet Tip Temperature (°C), Average Pipe Temperature (°C), Flow Rate (mL/s), Volume (mL)—float: four bytes Pump status (ON/OFF), heater status (ON/OFF), valve status (ON/OFF): inlet, sample 1, sample 2, cleaning, disposal—int: four bytes
Data Size	Data size frequency: 1 Hz Data packet size per second: 44 B Data packet size per sampling cycle: 510 KB

**Table 3 sensors-24-06107-t003:** Comparison of CFU/mL for Bacillus Subtilis culture, MilliQ water, and control.

Replicates	CFU/mL = Number of Colonies × Dilution Factor/Volume of Culture Plate		
	*Bacillus subtilis*	MilliQ Water after Cleaning and Sterilisation	MilliQ Water (Control)
1	1.59 × 10^9^	0	0
2	3.58 × 10^9^	0	0
3	6.15 × 10^7^ *	0	0

* a dilution factor of 10^4^ was used for culture plate replicate 3, while for the other two replicates (1 and 2), a dilution factor of 10^6^ was used.

## Data Availability

The source code for uploading to the ESP32S3 developed board and Simulink model and results can be found at https://github.com/miracleisraelnazarious/APLS (accessed on 9 September 2024).

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
