# Peer review of "Autonomous Planetary Liquid Sampler (APLS) for In Situ Sample Acquisition and Handling from Liquid Environments"

_sensors, 2024, doi:10.3390/s24186107_

Round 1

Reviewer 1 Report

Comments and Suggestions for Authors

The article describes the need to develop autonomous technologies for liquid sampling in hard-to-reach environments such as rivers, oceans, water storage tanks, and other similar settings.

In the "Introduction" section, several other liquid sampling technologies were mentioned; however, a discussion of the limitations of these technologies was not presented. This point is relevant so that in the "Results" and "Conclusions" sections, it can be demonstrated in which aspects the proposed APLS stands out or presents limitations compared to existing systems for sample collection in aquatic environments.

Although the article presents the experimental results in detail, the discussion of the implications of these results could be further elaborated. For example, the limitations of the prototype, such as its reliability in different environmental conditions or the challenges encountered during testing, were not mentioned.

The information contained in the supplementary material could be included in the "Results" and "Methods" sections of the main article. The comparisons between the experimental results and the simulations performed in Simulink are relevant to demonstrate the accuracy of the model in predicting the prototype's performance, which is a key point for validating the design and justifying the use of computational modeling.

Comments on the Quality of English Language

There are many words in British English; please confirm if they are correct.

sterilisable, characterisation, characterise, utilising, utilise

Author Response

Response to Reviewer 1:

The article describes the need to develop autonomous technologies for liquid sampling in hard-to-reach environments such as rivers, oceans, water storage tanks, and other similar settings.

Answer: Thank you for your valuable comments and insights. It helped us to resolve some of the gaps in the paper and make it more readable and useful to the community.

In the "Introduction" section, several other liquid sampling technologies were mentioned; however, a discussion of the limitations of these technologies was not presented. This point is relevant so that in the "Results" and "Conclusions" sections, it can be demonstrated in which aspects the proposed APLS stands out or presents limitations compared to existing systems for sample collection in aquatic environments.

Answer: Thank you for highlighting this need for a discussion. We have included a summary of the limitations of the state-of-the-art liquid samplers and presented the unique advantages of APLS in lines 79-83.

Although the article presents the experimental results in detail, the discussion of the implications of these results could be further elaborated. For example, the limitations of the prototype, such as its reliability in different environmental conditions or the challenges encountered during testing, were not mentioned.

Answer: Thank you for raising this concern. We agree with you that a suitable discussion is required elaborating the implications of the results. We have included that in lines 391-412 under subsection “3.5 Limitations and potential solutions”.

The information contained in the supplementary material could be included in the "Results" and "Methods" sections of the main article. The comparisons between the experimental results and the simulations performed in Simulink are relevant to demonstrate the accuracy of the model in predicting the prototype's performance, which is a key point for validating the design and justifying the use of computational modeling.

Answer: Thank you for addressing this. We agree with you that bringing the simulation results to the main paper will be better. Please find relevant modification in the main article pertaining to this in the Methods and Results section.

Comments on the Quality of English Language

There are many words in British English; please confirm if they are correct.

sterilisable, characterisation, characterise, utilising, utilise

Answer: Throughout the manuscript we have used British English scheme for writing. We will follow the advice of the Editors with this matter regarding the language preference. Thank you.

Reviewer 2 Report

Comments and Suggestions for Authors

I find the work interesting, as it focuses on improving a device for water sampling and quality testing. The solution proposed by the authors in the paper integrates several important functions for the autonomous operation of such a device. The authors have provided a clear introduction and literature review related to the topic of the paper, and presented the device's structure, its advantages, and the test results. I certainly believe there is room for improvement, and in line with that, I provide some comments and ask questions below, to which I would kindly ask the authors to respond:

Would the authors be so kind as to provide the reasons why the gear and screw pumps were eliminated?

I am not sure I understood what material the working elements of the selected pumps are made of—is it metal, ceramic, or plastic?

Metal without adequate protection, when in contact with liquids containing water or aggressive acids, will be attacked by corrosion. If corrosive particles enter the sample during system operation, they will contaminate it, interfere with the sensors and valves, and could potentially damage some system components.

Vane pumps have a very limited range of viscosity of the liquids they can work with and are often prone to cracking of the impeller due to hydraulic shock.

Piston pumps are probably the best choice if durability, longevity, and efficiency are required. Maybe authors should also include criteria of efficiency because I'm sure that the energy consumption spent by the pump during operation also has an impact on the overall performance of the device.

The authors state that the Inlet or suction filter should filter particles below 1 mm (row 133). Initially, I think that the authors have not adequately expressed themselves, and they must have meant particles above 1 mm. Furthermore, I ask that the material from which the filter is made be specified, as well as whether the solution for cleaning the filter has been implemented if it is submerged in water for a prolonged period. I am asking this question because, based on our experience, we know that in real-world conditions, materials tend to accumulate and/or multiply (if biological organisms are present), on these types of filters, resulting in their clogging, and subsequently, the pump's failure due to the difficulty in operation caused by the clogged suction filter.

In Table 1, the pattern 'Temperature Range of Electronics' appears three times! Why?

In hydraulics, shock waves are mitigated using hydraulic dampers or expansion tanks, which absorb the liquid's pressure oscillations during valve opening. Did the authors already test this option or see it from another reason that it was not suitable, so they decided to go with non-slam check valves? Have they even tested their proposed solution in the conditions of hydraulic shock? If the device is used in such an environment, this could be crucial for proper device operation.

Author Response

Response to Reviewer 2:

I find the work interesting, as it focuses on improving a device for water sampling and quality testing. The solution proposed by the authors in the paper integrates several important functions for the autonomous operation of such a device. The authors have provided a clear introduction and literature review related to the topic of the paper, and presented the device's structure, its advantages, and the test results. I certainly believe there is room for improvement, and in line with that, I provide some comments and ask questions below, to which I would kindly ask the authors to respond:

Answer: Thank you for your valuable comments and insights. It helped us to resolve some of the gaps in the paper and make it more readable and useful to the community.

Would the authors be so kind as to provide the reasons why the gear and screw pumps were eliminated?

Answer: Thank you for raising a good point. We have included our reasoning for omitting gear and screw pumps at lines 126-127.

I am not sure I understood what material the working elements of the selected pumps are made of—is it metal, ceramic, or plastic?

Answer: The current version uses a pump whose working elements are predominantly plastic. This has been clarified in line 212 and the need for metallic parts for the future versions has been justified in the newly added subsection “3.5 Limitations and potential solutions” in line 407.

Metal without adequate protection, when in contact with liquids containing water or aggressive acids, will be attacked by corrosion. If corrosive particles enter the sample during system operation, they will contaminate it, interfere with the sensors and valves, and could potentially damage some system components.

Answer: Thank you for your insight on the corrosion issue. We totally agree with you and added a clarification in lines 393-397.

Vane pumps have a very limited range of viscosity of the liquids they can work with and are often prone to cracking of the impeller due to hydraulic shock.

Piston pumps are probably the best choice if durability, longevity, and efficiency are required. Maybe authors should also include criteria of efficiency because I'm sure that the energy consumption spent by the pump during operation also has an impact on the overall performance of the device.

Answer: Thank you for your suggestion. We agree that the energy efficiency is one of the driving factors in our consideration for a suitable pump. We have mentioned that in lines 116 and 128.

The authors state that the Inlet or suction filter should filter particles below 1 mm (row 133). Initially, I think that the authors have not adequately expressed themselves, and they must have meant particles above 1 mm. Furthermore, I ask that the material from which the filter is made be specified, as well as whether the solution for cleaning the filter has been implemented if it is submerged in water for a prolonged period. I am asking this question because, based on our experience, we know that in real-world conditions, materials tend to accumulate and/or multiply (if biological organisms are present), on these types of filters, resulting in their clogging, and subsequently, the pump's failure due to the difficulty in operation caused by the clogged suction filter.

Answer: Thank you for a beautiful insight on the issue of clogging. We have addressed to limit this issue in our current design by using a copper inlet filter that inhibit bacterial growth. We have explicitly mentioned it in line 228 and lines 230-232 with a suitable reference. We have also corrected the typo from “below 1 mm” to “above 1 mm” in line 139.

In Table 1, the pattern 'Temperature Range of Electronics' appears three times! Why?

Answer: Thank you for pointing it out. It was a mistake from our side. It has now been corrected.

In hydraulics, shock waves are mitigated using hydraulic dampers or expansion tanks, which absorb the liquid's pressure oscillations during valve opening. Did the authors already test this option or see it from another reason that it was not suitable, so they decided to go with non-slam check valves? Have they even tested their proposed solution in the conditions of hydraulic shock? If the device is used in such an environment, this could be crucial for proper device operation.

Answer: Thank you for your suggestion. At this time, we are proposing different solutions to best manage hydraulic shock waves. This will be implemented in the future design of the APLS system. We have not tested any of the proposed solution in the conditions of hydraulic shock yet.  We have added your suggestion to the list of possible solutions we will add to the system in the future in line 196.

Round 2

Reviewer 1 Report

Comments and Suggestions for Authors

The manuscript was improved, and all suggestions were accepted. The discussions about the limitations of the technology were added to the final manuscript. The modification related to moving the simulation results to the main paper enriched the document.